# Relationship between tongue pressure and dysphagia diet in patients with acute stroke

**Masahiro Nakamori** [1,2]*, **Kenichi Ishikawa**[1,2], **Eiji Imamura**[2], **Haruna Yamamoto**[3], **Keiko Kimura**[3], **Tomoko Ayukawa**[4], **Tatsuya Mizoue**[5], **Shinichi Wakabayashi**[5]

**1** Department of Clinical Neuroscience and Therapeutics, Hiroshima University Graduate School of Biomedical and Health Sciences, Hiroshima, Japan, **2** Department of Neurology, Suiseikai Kajikawa Hospital, Hiroshima, Japan, **3** Department of Nursing, Suiseikai Kajikawa Hospital, Hiroshima, Japan, **4** Department of Rehabilitation, Suiseikai Kajikawa Hospital, Hiroshima, Japan, **5** Department of Neurosurgery, Suiseikai Kajikawa Hospital, Hiroshima, Japan

* mnakamori1@gmail.com

**Data Availability Statement:** All relevant data are within the manuscript and its Supporting information files.

## Abstract

A dysphagia diet is important for patients with stroke to help manage their nutritional state and prevent aspiration pneumonia. Tongue pressure measurement is a simple, non-invasive, and objective method for diagnosing dysphagia. We hypothesized that tongue pressure may be useful in making a choice of diet for patients with acute stroke. Using balloon-type equipment, tongue pressure was measured in 80 patients with acute stroke. On admission, a multidisciplinary swallowing team including doctors, nurses, speech therapists, and management dietitians evaluated and decided on the possibility of oral intake and diet form; the tongue pressure was unknown to the team. Diet form was defined and classified as dysphagia diet Codes 0 to 4 and normal form (Code 5 in this study) according to the 2013 Japanese Dysphagia Diet Criteria. In multivariate analysis, only tongue pressure was significantly associated with the dysphagia diet form (p<0.001). Receiver operating characteristic analyses revealed that the optimal cutoff tongue pressure for predicting diet Codes 1, 2, 3, 4, and 5 was 3.6 (p<0.001, area under the curve [AUC] = 0.997), 9.6 (p<0.001, AUC = 0.973), 12.8 (p<0.001, AUC = 0.963), 16.5 (p<0.001, AUC = 0.979), and 17.3 kPa (p<0.001, AUC = 0.982), respectively. Tongue pressure is one of the sensitive indicators for choosing dysphagia diet forms in patients with acute stroke. A combination of simple modalities will increase the accuracy of the swallowing assessment and choice of the diet form.

## Introduction

For patients with stroke, dysphagia may lead to aspiration pneumonia and malnutrition. Prevention of aspiration pneumonia and nutrition management decreases the duration of hospitalization and mortality [1–3]. Thus, it is essential to serve a dysphagia diet to manage the patient's nutritional state and prevent aspiration pneumonia. In particular, the multidisciplinary swallowing team approach decreases the onset of pneumonia in patients with acute stroke [4].

**Funding:** The authors received no specific funding for this work.

**Competing interests:** The authors have declared that no competing interests exist.

Videofluoroscopic and videoendoscopic examinations are the most accurate instrumental assessment tools for evaluating swallowing function. However, some patients cannot be evaluated fully due to their conditions or the unavailability of such tools. Thus, bedside screening tests, such as the repetitive saliva swallowing test and water swallowing test, are simple and non-invasive tests for assessing swallowing dysfunction [5, 6]. However, these tests are not accurate. Other auxiliary methods such as tongue ultrasonography and tongue pressure measurement are useful for evaluating dysphagia [7–9].

Concerning the prevention of pneumonia and management of nutrition, the possibility of oral intake and dysphagia diet should be chosen after assessing the swallowing function. However, there are few objective, simple, and non-invasive methods. It is very important to find an objective scale for choosing a suitable dysphagia diet form in clinical practice. In this study on acute stroke care, we hypothesized that tongue pressure would be a useful indicator for choosing the diet form for patients with acute stroke. We investigated the association between tongue pressure and dysphagia diet form and searched for the suitable tongue pressure for choosing each diet form.

## Materials and methods

### Ethics approval

This study had a prospective cohort design. This study was approved by the ethics committee of Suiseikai Kajikawa Hospital and was performed according to the national government's guidelines based on the Helsinki Declaration of 1964. Written informed consent was obtained from all the patients or their relatives. All assessors were blinded to the data for analyses.

### Subjects

All consecutive patients with acute ischemic or hemorrhagic stroke who were admitted to Suiseikai Kajikawa Hospital within 1 week of disease onset between June 1, 2015, and August 31, 2015 were included in this cohort study. Patients who were under 20 years old, did not provide consent (for patients who could not provide consent, consent was obtained from their relatives), were in coma (best eye response score on the Glasgow coma scale of 1), underwent craniotomy, were on mechanical ventilation, or diagnosed with aspiration pneumonia in the hospital were excluded. Aspiration pneumonia was diagnosed according to the criteria of the Centers for Disease Control and Prevention [10]. We monitored patients for signs of pneumonia for 30 days after admission. If the patients were discharged before the 30th day, then they were monitored until the day of discharge.

### Data acquisition

The stroke subtype was determined according to the Trial of Org 10172 in Acute Stroke Treatment classification [11]. Stroke severity was evaluated using the National Institutes of Health Stroke Scale (NIHSS) score [12].

On admission, a multidisciplinary swallowing team including doctors, nurses, speech therapists, and management dietitians evaluated and decided on the possibility of oral intake and diet form; the tongue pressure was unknown to the team. Diet form was defined and classified as dysphagia diet Codes 0 to 4 and normal form according to the 2013 Japanese Dysphagia Diet Criteria [13]. In this code, dysphagia diet Code 0 refers to a soft, homogeneous diet, with a low adherability, high coherence, less water separation, and which can form a suitable bolus. Code 1 refers to a homogeneous diet, with less water separation, and which can form a suitable bolus. This does not require the ability to chew. It is often called jelly, pudding, or mousse

food. Code 2 refers to a diet that readily forms a suitable bolus via simple oral movement. The tongue is required to press the hard palate during transportation. It is often called blender, puree, or paste food. Code 3 refers to a solid diet that can be crushed without teeth or prosthodontics and can easily form a bolus. Owing to the need for mouth movements, less water separation, and moderate coherence, this food is difficult to separate during its passage via the pharynx. It is often referred to as soft food. Code 4 refers to a diet that needs adjustment of the material and cooking method to prevent aspiration or suffocation. It is not too hard to form, come apart, and it has a high adhesion. It is soft enough to be cut with a spoon or chopsticks. It does not require teeth or prosthodontics but can be crushed with both alveolar ridges. It is difficult to crush food if only the tongue is pressed onto the hard palate. In this study, we defined a diet with a normal form as Code 5. Swallowing was evaluated using the Food Intake Level Scale (FILS) [14]. The FILS is a 10-point observer-rating scale that measures the severity of swallowing dysfunction. Its convergent validity and intra-rater and inter-rater reliabilities have been established using the Functional Oral Intake Scale.

On the same day, clinical technicians measured the tongue pressure using balloon-type equipment (TPM-01; JMS Co. Ltd., Hiroshima, Japan) independently. The balloon-type equipment consists of a disposable oral probe, an infusion tube as a connector, and a recording device. The patients were seated during the tongue pressure measurement and were asked to put the balloon in their mouths. They held the pipe at the midpoint of their central teeth. They were asked to maintain this position while the examiners adjusted the probe and confirmed the correct position. The patients were then asked to raise their tongue and push the balloon against their palate using maximum efforts for seven seconds, according to previous reports [15, 16]. This measurement was performed three times; the patients rested for 30 seconds and rinsed their mouth between each measurement. The maximum value of the three measurements for each patient was considered as the tongue pressure.

The reliability of intraindividual measurements has been reported previously [17]. We reconfirmed the reliability of the measurements in this study. Measurements were taken repeatedly for ten days in normal subjects, and the resulting coefficient of variation was 5.64%.

## Statistical analysis

Data are expressed as median (minimum, 25th percentile, 75th percentile, maximum) for continuous variables (age, body mass index, NIHSS score, FILS, diet form, and tongue pressure) and as frequencies (percentages) for categorical variables (stroke subtypes and histories of comorbidities). Statistical analysis was performed using JMP 15 statistical software (SAS Institute Inc., Cary, NC, USA). We calculated the required sample size based on our previous study on tongue pressure in acute stroke [7]. In the previous study, tongue pressure was compared using the modified Mann Assessment of Swallowing Ability (MASA) score, which is an established bedside assessment tool that indicates the risk of developing swallowing dysfunction. A modified MASA score of <95 suggests swallowing dysfunction. The difference in the tongue pressure between the modified MASA score <95 and ≥95 groups was investigated, and the effect size was strong. This parameter was used to calculate the study sample size. Using an alpha level of 0.05 and a power of 0.80, 56 participants were required to participate in this study. To evaluate the association between tongue pressure and dysphagia diet form, the baseline data of the patients were analyzed, and two-step strategies were used to assess the significant importance of variables in the association between tongue pressure and FILS/dysphagia diet form using a least square linear regression analysis. First, a univariate analysis was performed, and factors with p <0.05 were selected. In a multi-factorial analysis, least linear regression analyses were performed with the selected factors. The analysis of co-variance was used to

compare the mean tongue pressure between the dysphagia diet groups. Moreover, receiver operating characteristic (ROC) analyses were performed to determine the tongue pressure to predict suitable diet forms. Statistical significance was set at p <0.05.

## Results

A total of 105 patients were eligible for this study. No patient was below 20 years old. Eleven patients or their relatives did not provide consent, two patients underwent craniotomy, and one patient was on mechanical ventilation. Among the remaining patients, eight patients were not evaluated because of a poor general condition, and three patients were diagnosed with aspiration pneumonia. Therefore, 80 patients were included in the study. A flowchart of the inclusion and exclusion criteria is shown in Fig 1. Data of the subjects' backgrounds and characteristics are shown in Table 1. The mean tongue pressure was 21.5±14.1 kPa.

The associations between the factors listed in Table 1 (except for FILS and dysphagia diet form because they are response variables) and tongue pressure at admission were evaluated. In univariate analysis, age, sex, body mass index, diabetes mellitus, dyslipidemia, and NIHSS score were associated with tongue pressure. In multivariate analysis, age and NIHSS score were significant independent factors for tongue pressure (Table 2).

Next, the associations between the factors listed in Table 1 and FILS/dysphagia diet form were evaluated. Regarding FILS in univariate analysis, age, sex, dyslipidemia, NIHSS score, and tongue pressure were associated with FILS. In multivariate analysis, NIHSS score and tongue pressure were significant independent factors for FILS (Table 3A). Regarding dysphagia diet form, in univariate analysis, age, sex, body mass index, NIHSS score, and tongue pressure were associated with dysphagia diet form. In multivariate analysis, only tongue pressure was significantly associated with the dysphagia diet form (Table 3B).

The mean tongue pressures of the dysphagia diet groups were compared (Fig 2). After adjusting for age and NIHSS score, which were associated with tongue pressure, the mean tongue pressure increased significantly as the dysphagia diet code increased.

In the analysis of co-variance, the mean tongue pressures increased significantly as the dysphagia diet code increased.

Because we revealed the association between FILS/dysphagia diet form and tongue pressure (Table 3), ROC analyses were performed to determine the tongue pressure that can be used to predict a suitable dysphagia diet. FILS measures swallowing ability without considering the diet form. However, in a real clinical setting, it is important to know whether the patient can take food completely orally. A state of complete oral intake corresponds to FILS ≥7; therefore, we investigated the optimal cutoff tongue pressure for predicting a FILS of ≥7. The cut-off was 9.6 kPa ($\chi^2$ = 46.83, p<0.001, sensitivity 100.0%, specificity 92.8%, AUC = 0.991) (Fig 3A). Next, the optimal cut-off tongue pressure for predicting suitable dysphagia diet forms using the diet form code was investigated. The optimal cutoff tongue pressures to predict Codes 1, 2, 3, 4, and 5 were 3.6 kPa ($\chi^2$ = 53.12, p<0.001, sensitivity 100.0%, specificity 97.1%, AUC = 0.997), 9.6 kPa ($\chi^2$ = 50.87, p<0.001, sensitivity 92.9%, specificity 95.5%, AUC = 0.973), 12.8 kPa ($\chi^2$ = 55.85, p<0.001, sensitivity 89.5%, specificity 90.2%, AUC = 0.963), 16.5 kPa ($\chi^2$ = 74.90, p<0.001, sensitivity 96.4%, specificity 88.5%, AUC = 0.979), and 17.3 kPa ($\chi^2$ = 83.34, p<0.001, sensitivity 95.7%, specificity 99.9%, AUC = 0.982), respectively. The ROC curves are shown in Fig 3B–3F.

## Discussion

In this study, we measured the tongue pressure in patients with acute stroke and found that it was significantly associated with the FILS and dysphagia diet forms. The optimal cutoff tongue

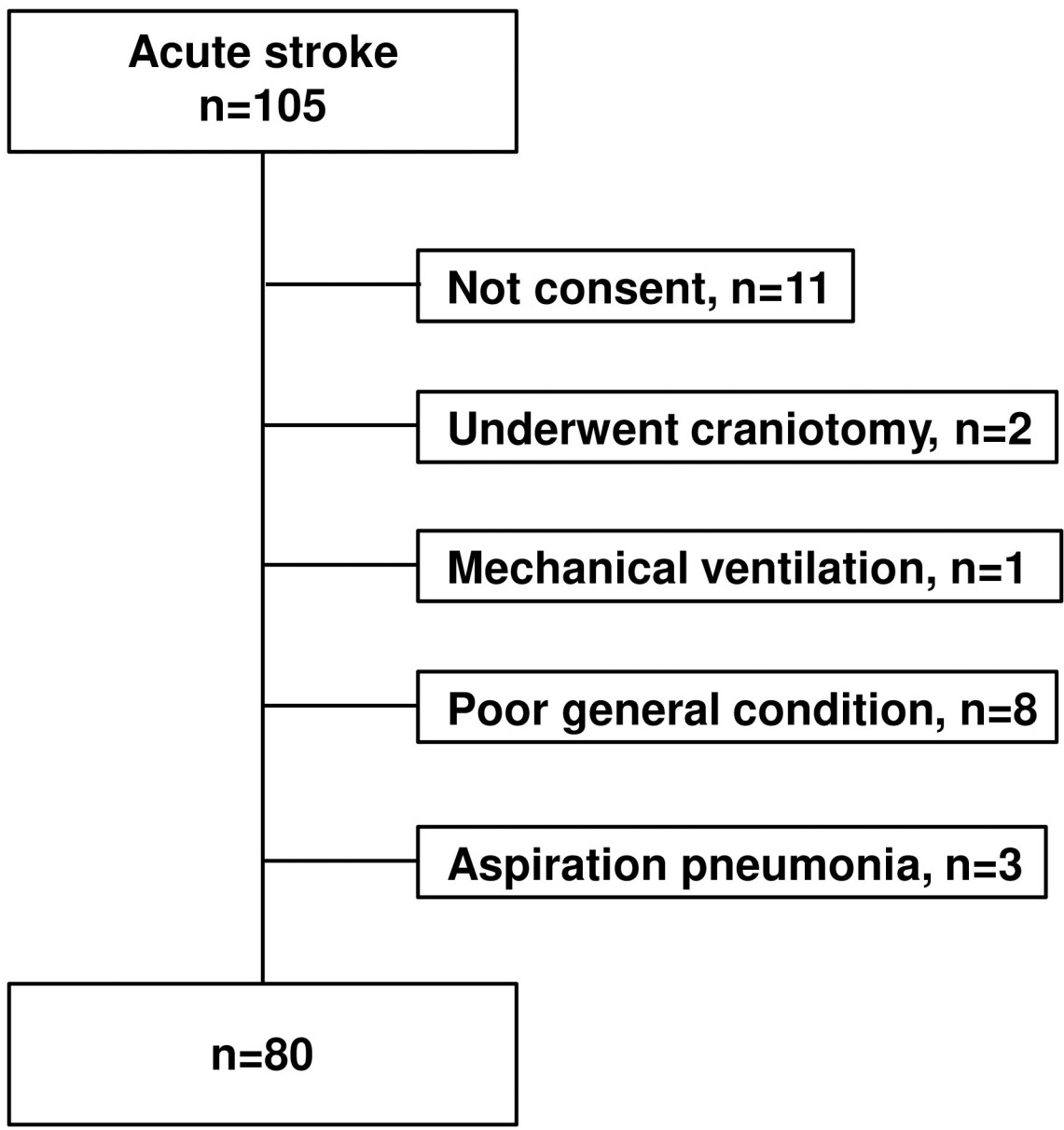

**Fig 1. Flow chart of the inclusion and exclusion criteria.**

pressure for predicting a FILS ≥7 (the FILS for patients to receive food orally completely without intravenous nutrition) was 9.6 kPa. The mean tongue pressure increased significantly with an increase in the dysphagia diet code. The ROC analyses revealed that the optimal cutoff tongue pressure for predicting diet form Codes 1, 2, 3, 4, and 5 was 3.6, 9.6, 12.8, 16.5, and 17.3 kPa (p<0.001, AUC = 0.982), respectively. These results provide objective, practical, and useful indications for choosing a suitable dysphagia diet.

The tongue has a significant role in swallowing [18]. The tongue forms a bolus and sends the bolus to the pharynx, where the epiglottis moves downward and closes the larynx. Tongue pressure is a quantitative measurement of the tongue biomechanics during swallowing [16, 19, 20]. A lower tongue pressure prevents bolus control and increases oral residues; these increase

**Table 1. Patient characteristics (n = 80).**

| Factors | |
|---|---|
| Age, years | 75 (53, 67, 84, 98) |
| Women | 41 (51.3) |
| Body mass index, kg/m$^2$ | 22.2 (13.3, 19.8, 25.5, 32.9) |
| Stroke Subtypes | |
| ATBI | 18 (22.50) |
| CEI | 12 (15.00) |
| LI | 13 (16.25) |
| Others | 24 (30.00) |
| ICH | 13 (16.25) |
| Hypertension | 61 (76.25) |
| Diabetes mellitus | 22 (27.50) |
| Dyslipidemia | 30 (37.50) |
| Atrial fibrillation | 14 (17.50) |
| NIHSS score | 4 (0, 2, 9.75, 36) |
| FILS | 9 (1, 7, 10, 10) |
| diet form | 5 (0, 3, 5, 5) |
| Tongue pressure, kPa | 21.7 (0, 12.1, 31.9, 53.1) |

Data are expressed as median (minimum, 25$^{th}$ percentile, 75$^{th}$ percentile, maximum) for continuous variables (age, body mass index, NIHSS score, FILS, diet form, and tongue pressure), and frequencies (percentages) for categorical variables (stroke subtypes and histories of comorbidities).

BMI, body mass index; ATBI, atherothrombotic brain infarction; CEI, cardiogenic embolism infarction; LI, lacunar infarction; ICH, intracerebral hemorrhage; NIHSS, National Institutes of Health Stroke Scale; FILS, Food Intake Level Scale.

the risk of aspiration. Additionally, a lower tongue pressure is related to inadequate closing, which can be a risk of aspiration during swallowing. Tongue pressure is associated with oro-pharyngeal residues visualized during videofluoroscopic examination [21]. It is reported that with a water swallowing test, tongue pressure is lower in patients with dysphagia than in those without [22]. Tongue pressure measurement is a non-invasive and straightforward tool with significant advantages. Moreover, the relationship between tongue pressure and videofluoro-scopic examination, which is the gold standard method, has been established.

Among healthy Japanese subjects, the standard tongue pressure, measured using the same device, reduced with increasing age (41.7±9.7 kPa in subjects in their 20s and 31.9±8.9 kPa in subjects in their 70s) [19]. In addition, frail elderly Japanese subjects had a tongue pressure of 18.0±12.0 kPa [20, 23]. Moreover, the tongue pressures were lower than those of healthy subjects [19]. Our institution reported that a tongue pressure <21.6 kPa suggested swallowing dysfunction, in which the modified MASA score was less than 95 [7]. This study revealed that the indication for a normal-form diet was 17.3 kPa, which was consistent with previous reports.

There are several standards for dysphagia diet. The diet differs among regions and countries because of the different food cultures. Thus, the definition and form of dysphagia diet foods might differ among countries. The 2013 Japanese Dysphagia Diet Criteria were introduced for a consensus of dysphagia diet form for many hospitals and institutions [13]. In addition, Code 0 was divided into jelly and thickened. In this study, tongue pressure was strongly associated with diet forms, and ROC analyses revealed the optimal tongue pressure for choosing the diet form. In patients with acute stroke, tongue pressure is useful in choosing the dysphagia diet

**Table 2. Factors influencing tongue pressure.**

| Factors | Univariate analysis | Multivariate analysis | | |
|---|---|---|---|---|
| | p value | Regression coefficient | 95% CI | p value |
| Age | <0.001 | -0.429 | -0.693–-0.164 | 0.002 |
| Sex | <0.001 | 1.740 | 0.905–4.384 | 0.194 |
| Body mass index | 0.003 | 0.097 | -0.593–0.787 | 0.779 |
| Stroke Subtypes | 0.087 | | | |
| Hypertension | 0.815 | | | |
| Diabetes mellitus | 0.032 | -0.786 | -3,551–1.979 | 0.573 |
| Dyslipidemia | 0.005 | 0.162 | -2.519–2.844 | 0.904 |
| Atrial fibrillation | 0.299 | | | |
| NIHSS score | <0.001 | -0.712 | -1.043–-0.380 | <0.001 |

Regarding the categorical variables, the reference for sex was female, and the reference for hypertension, diabetes mellitus, dyslipidemia, and atrial fibrillation were normal controls.

CI, confidence interval; NIHSS, National Institutes of Health Stroke Scale.

**Table 3. Factors influencing the Food Intake Level Scale and dysphagia diet form.**

| Factors | Univariate analysis | Multivariate analysis | | |
|---|---|---|---|---|
| | p value | Regression coefficient | 95% CI | p value |
| **A. Factors influencing the Food Intake Level Scale** | | | | |
| Age | <0.001 | -0.001 | -0.047–0.045 | 0.957 |
| Sex | 0.006 | 0.093 | -0.342–0.528 | 0.672 |
| Body mass index | 0.114 | | | |
| Stroke Subtypes | 0.138 | | | |
| Hypertension | 0.947 | | | |
| Diabetes mellitus | 0.220 | | | |
| Dyslipidemia | 0.001 | -0.094 | -0.512–0.324 | 0.656 |
| Atrial fibrillation | 0.457 | | | |
| NIHSS score | <0.001 | -0.146 | -0.207–-0.084 | <0.001 |
| Tongue pressure | <0.001 | 0.110 | 0.072–0.149 | <0.001 |
| **B. Factors influencing the dysphagia diet form** | | | | |
| Age | <0.001 | -0.013 | -0.090–0.064 | 0.724 |
| Sex | 0.034 | 0.305 | -0.391–1.012 | 0.398 |
| Body mass index | 0.023 | 0.022 | -0.142–0.178 | 0.788 |
| Stroke Subtypes | 0.324 | | | |
| Hypertension | 0.128 | | | |
| Diabetes mellitus | 0.731 | | | |
| Dyslipidemia | 0.068 | | | |
| Atrial fibrillation | 0.248 | | | |
| NIHSS score | <0.001 | -0.058 | -0.148–0.026 | 0.203 |
| Tongue pressure | <0.001 | 0.448 | 0.307–0.622 | <0.001 |

Regarding the categorical variables, the reference for sex was female, and the reference for hypertension, diabetes mellitus, dyslipidemia, and atrial fibrillation were normal controls.

CI, confidence interval; NIHSS, National Institutes of Health Stroke Scale.

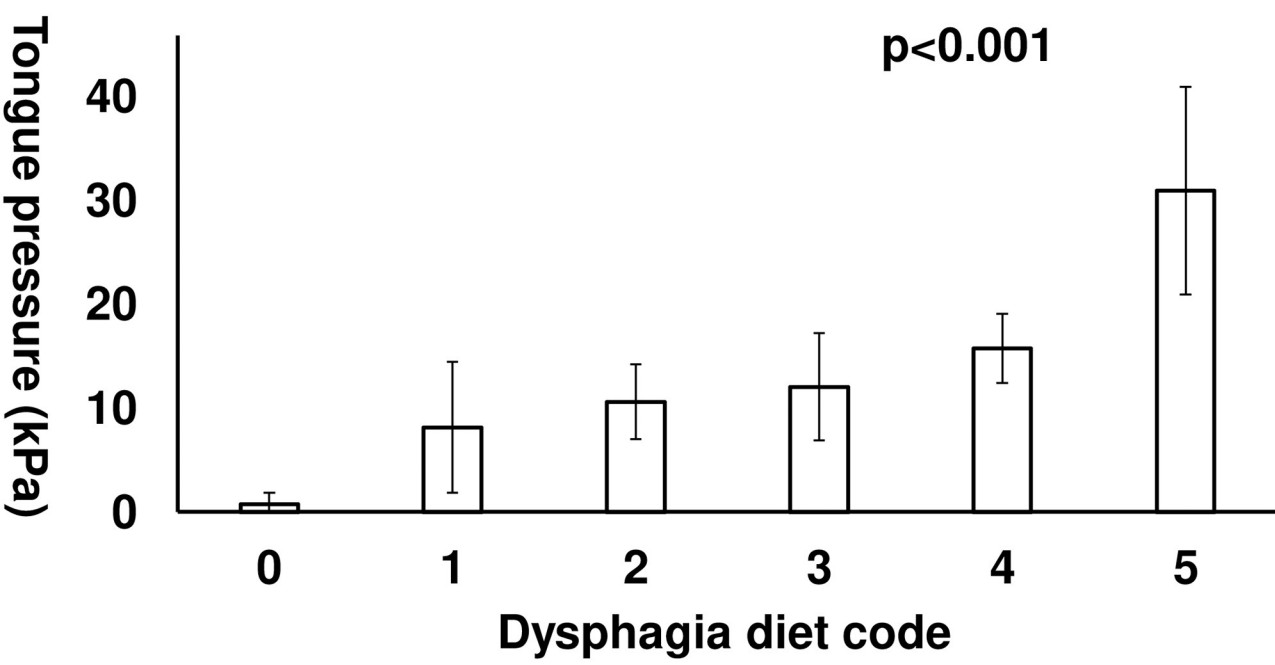

**Fig 2. Tongue pressure by dysphagia diet code.**

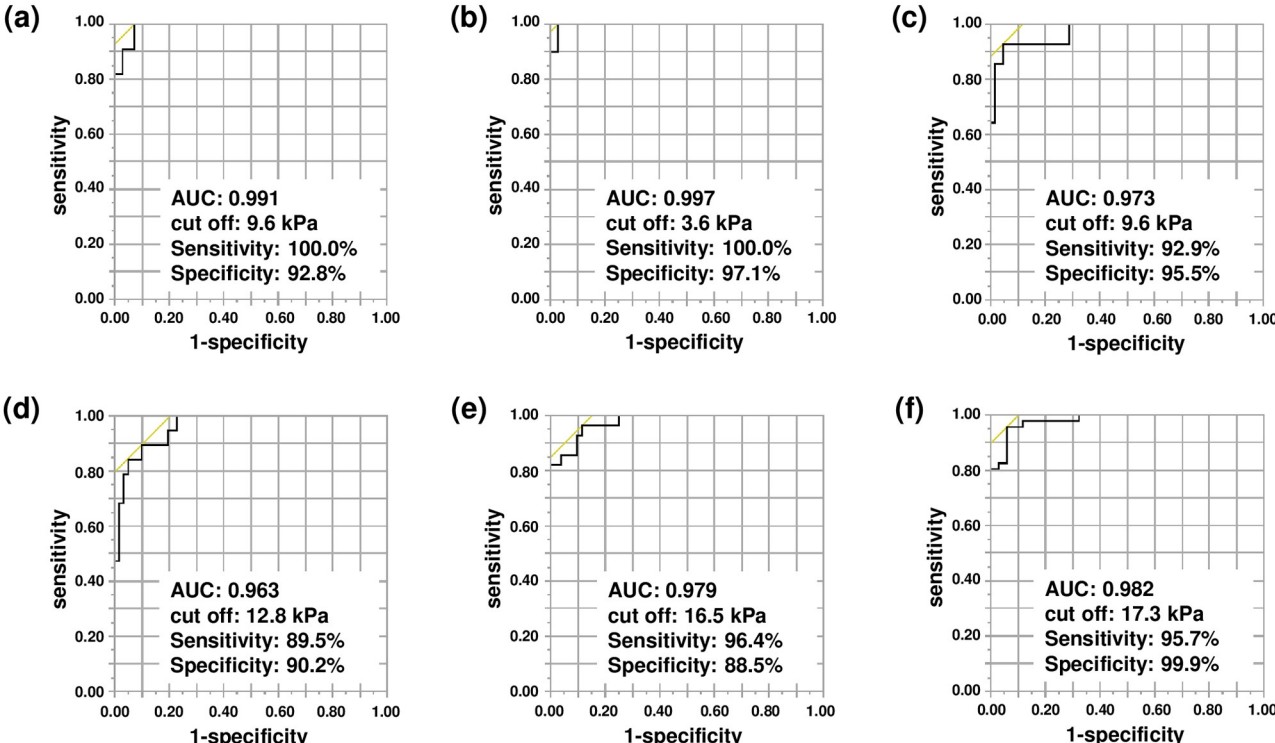

**Fig 3. Receiver operating characteristic curve of tongue pressure for predicting dysphagia diet form.** FILS ≥7 (A), Code ≥1 (2013 Japanese Dysphagia Diet Criteria) (B), Code ≥2 (C), Code ≥3 (D), Code ≥4 (E), and Code 5 (F).

form. In addition, these results indicate that the dysphagia diet is strongly dependent on tongue function. However, to prevent aspiration pneumonia, there are some important factors, such as cough function. The cough reflex is an essential function for protecting and clearing materials that are aspirated into the airway. It has also been reported that the cough test is useful for screening silent aspiration [24]. The decision of fasting should be done by such an assessment, but not by tongue pressure.

In clinical practice, it is important for a multidisciplinary swallowing team to diagnose dysphagia as early as possible to start managing nutrition and preventing aspiration pneumonia. Although we reconfirmed the reliability of tongue pressure measurement, it does not require special skills. In this study, tongue pressure was measured on admission—this is recommendable.

This study had limitations. First, it was performed in a single institution and might have sources of bias. The sample size was small. More than half of the patients had mild stroke, which could prevent the generalizability of the results to a population with severe stroke. It would be difficult to measure the tongue pressure of patients with severe stroke accurately. Future multicenter research would be needed to eliminate the effects of bias. Second, tongue pressure cannot reflect all functions for swallowing. Videofluoroscopic and videoendoscopy examinations are the gold standards for evaluating dysphagia; however, they have limitations, such as exposure to radiation and difficulties to be performed in disabled patients [25]. In addition, it is impossible to perform these examinations in all patients with acute stroke. Thus, the swallowing assessment and decision for the diet form must be accurately performed using a combination of simple, non-invasive, and bedside modalities such as ultrasonography and cough test [9].

## Conclusions

Tongue pressure is a sensitive and useful indicator for choosing the dysphagia diet form in patients with acute stroke. As a bedside assessment tool, tongue pressure measurement helps in nutrition management and evaluation of the swallowing function. The combination of simple modalities will increase the accuracy of the swallowing assessment and choice of the diet form.

## Supporting information

**S1 Data. All relevant data of the study.**
(XLSX)

## Acknowledgments

We would like to sincerely thank the staff in the Department of Nutrition at the Suiseikai Kajikawa Hospital for their technical assistance.

## Author Contributions

**Conceptualization:** Masahiro Nakamori, Haruna Yamamoto, Keiko Kimura, Tomoko Ayukawa, Shinichi Wakabayashi.

**Data curation:** Masahiro Nakamori, Kenichi Ishikawa, Eiji Imamura, Haruna Yamamoto, Keiko Kimura, Tomoko Ayukawa.

**Formal analysis:** Masahiro Nakamori, Eiji Imamura, Haruna Yamamoto, Keiko Kimura.

**Investigation:** Masahiro Nakamori, Kenichi Ishikawa, Eiji Imamura, Haruna Yamamoto, Keiko Kimura, Tomoko Ayukawa.

**Methodology:** Masahiro Nakamori, Eiji Imamura, Haruna Yamamoto, Keiko Kimura, Tomoko Ayukawa.

**Project administration:** Masahiro Nakamori, Haruna Yamamoto, Keiko Kimura, Tomoko Ayukawa.

**Resources:** Masahiro Nakamori, Haruna Yamamoto, Keiko Kimura, Tomoko Ayukawa, Shinichi Wakabayashi.

**Supervision:** Eiji Imamura, Tatsuya Mizoue, Shinichi Wakabayashi.

**Validation:** Masahiro Nakamori, Eiji Imamura, Keiko Kimura, Tatsuya Mizoue, Shinichi Wakabayashi.

**Writing – original draft:** Masahiro Nakamori, Kenichi Ishikawa, Eiji Imamura.

**Writing – review & editing:** Masahiro Nakamori, Eiji Imamura, Tatsuya Mizoue, Shinichi Wakabayashi.

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
