## [Decision Letter · Decision Letter 0]

9 Apr 2021

PONE-D-21-05820

The relationship between tongue pressure and dysphasia diet in acute stroke patients

PLOS ONE

Dear Dr. Masahiro Nakamori,

Thank you for submitting your manuscript to PLOS ONE. After careful consideration, we feel that it has merit but does not fully meet PLOS ONE’s publication criteria as it currently stands. Therefore, we invite you to submit a revised version of the manuscript that addresses the points raised during the review process.

Although it is of interest, the reviewers have raised a number of points which we believe major modifications are necessary to improve the manuscript, taking into account the reviewers' remarks.

We look forward to receiving your revised manuscript.

Kind regards,

Wisit Cheungpasitporn, MD

Academic Editor

PLOS ONE

Journal Requirements:

2. Thank you for submitting the above manuscript to PLOS ONE. During our internal evaluation of the manuscript, we found significant text overlap between your submission and the following previously published work, of which you are an author.

- https://journals.plos.org/plosone/article?id=10.1371%2Fjournal.pone.0239590

Please revise the manuscript to rephrase the duplicated text, cite your sources, and provide details as to how the current manuscript advances on previous work. Please note that further consideration is dependent on the submission of a manuscript that addresses these concerns about the overlap in text with published work.

Reviewers' comments:

Reviewer's Responses to Questions

**Comments to the Author**

1. Is the manuscript technically sound, and do the data support the conclusions?

Reviewer #1: Partly

Reviewer #2: Yes

Reviewer #3: Yes

2. Has the statistical analysis been performed appropriately and rigorously? 

Reviewer #1: No

Reviewer #2: Yes

Reviewer #3: Yes

3. Have the authors made all data underlying the findings in their manuscript fully available?

Reviewer #1: Yes

Reviewer #2: No

Reviewer #3: Yes

4. Is the manuscript presented in an intelligible fashion and written in standard English?

Reviewer #1: Yes

Reviewer #2: Yes

Reviewer #3: Yes

5. Review Comments to the Author

Reviewer #1: Methods:

Should state the study design and duration of study.

Please add explanation on sample size determination; the parameters used in the calculation.

Please explain on sampling method in related to “consecutive acute stroke patients…”.

Should report median with interquartile range.

Revise the findings. Should state the outcome for each analysis. Tongue pressure influence the dysphagia diet or vice versa. Please explain.

The authors should control the confounder for comparison of mean tongue pressure for each group of dysphagia diet; ANCOVA instead of ANOVA.

Do all the assumptions of the test fulfilled? Please explain.

Please justify the use of ROC analysis.

Results:

Revise the findings.

Table 1: Delete All. Should write n=80 at the end of table title. Replace it with n(%). Delete n(%) written after factors. Should report median (IQR).

Table 2: Any reason run different model. What is predictor? Does it refer to regression coefficient? Please the results of categorical factors. Should state the reference.

Reviewer #2: This study by Nakamori et al evaluated if tongue pressure measurement is a sensitive indicator for determining diet form and swallowing capability among a sample of 80 patients hospitalized for acute stroke in a Japanese hospital. The authors found that tongue pressure measured on admission by a multidisciplinary team is significantly associated with diet form and swallowing. The manuscript is very well-written and describes an interesting research question with clear clinical implications. I have some comments about the presentation of the methods and results:

Methods:

1. Page 10 lines 144-146: Please describe the additional parameters used to calculate sample size (e.g., effect size based on prior literature)

2. Page 10 lines 148: More details about the multivariable model are important to state. Please explicitly state what the dependent variable is and how it was operationalized (continuous, categorical, etc). Along the same lines, what type of multivariable model was used (linear, logistic, etc)? What covariates did you consider in the model and how were they operationalized (Was NIHSS at admission and was it measured as a continuous variable? Same with BMI and comorbidities.)

Results:

3. Page 11 (Table 1): It would be more helpful to know the IQR when you present descriptive statistics (rather than minimum and maximum), especially for your key variables of interest (FILS, diet form). More information about distribution of the FILS, diet form, and tongue pressure, would be helpful to the reader since these are key variables in your study.

4. Page 13 (Table 2): Some comments on the presentation of data in Table 2. What is the reference group for categorical variables (sex, hypertension, diabetes, etc)? This should be clearly labeled in the table. Is the “predictor” column referring to beta coefficients?

5. Page 15 Figure 1: Please indicate that the bars are (95%?) confidence intervals

6. Page 15 lines 191-192: The mention of ROC analyses was a surprise as I was reading the article. ROC needs to be described in the methods section.

General comments:

7. I’m a little confused about the wording in the manuscript. Are you classifying FILS (swallowing capability) as another measure of diet form? For example, the caption for figure 2 only refers to diet form, even though panel A discusses FILS.

8. It may be useful to have information about training to conduct tongue pressure. Who performed it in the multidisciplinary team? Was there standardized training or is this part of routine clinical practice so this type of standardization is not needed? This would be useful if the reader wants to consider how they can translate these results to their own clinical practice.

Reviewer #3: Dear Dr. Masahiro Nakamori:

Thank you for allowing us to read your interesting manuscript. This study provides information about the tongue pressure, such as a non-invasive bedside assessment tool for deciding the diet form for acute stroke patients. It is a great effort, congratulations. Please consider the following comments.

In many parts of the manuscript appears the word “dysphasia” (9 times) including the tittle, which I think should be corrected to “dysphagia”.

It should be important that the abstract conclusion make it clear that tongue pressure is just one tool within other evaluations (such cough function) that must be made to patient to decide the type of diet, similar to the idea expressed in the extend conclusion.

More than 50% of the sample were minor strokes (NIHSS �5), which could be a limitation when extending the results to more serious strokes.

It should be important that you give a recommendation regarding the appropriate timing for the initiation of this post-stroke test (tongue pressure), depending on the moment in which you carried out these evaluations in the study.

Thanks,

6. PLOS authors have the option to publish the peer review history of their article (what does this mean?). If published, this will include your full peer review and any attached files.

Reviewer #1: No

Reviewer #2: No

Reviewer #3: **Yes: **Vanessa Cano-Nigenda

---

## [Author Response · Author response to Decision Letter 0]

30 Apr 2021

Thank you for reviewing our work. We appreciate all your comments and suggestions. We have revised the manuscript accordingly. Our point-by-point responses are presented below.

Response to the Editor

Comment 1: Please ensure that your manuscript meets PLOS ONE's style requirements, including those for file naming. The PLOS ONE style templates can be found at https://journals.plos.org/plosone/s/file?id=wjVg/PLOSOne_formatting_sample_main_body.pdf and https://journals.plos.org/plosone/s/file?id=ba62/PLOSOne_formatting_sample_title_authors_affiliations.pdf

Response 1: We have ensured that the manuscript meets the style requirements of PLOS ONE.

Comment 2: Thank you for submitting the above manuscript to PLOS ONE. During our internal evaluation of the manuscript, we found significant text overlap between your submission and the following previously published work, of which you are an author.

- https://journals.plos.org/plosone/article?id=10.1371%2Fjournal.pone.0239590

Please revise the manuscript to rephrase the duplicated text, cite your sources, and provide details as to how the current manuscript advances on previous work. Please note that further consideration is dependent on the submission of a manuscript that addresses these concerns about the overlap in text with published work.

Response 2: As you pointed out, we have checked and revised the manuscript. We have also ensured that all text overlaps were avoided. Our revisions are highlighted in red font in the revised manuscript.

 

Response to the reviewers

Reviewer #1: 

Comment 1: Should state the study design and duration of study.

Response 1: Thank you for this suggestion. This study had a prospective cohort design. We monitored patients for signs of pneumonia for 30 days after admission. If the patients were discharged before the 30th day, then they were monitored until the day of discharge. We have added this information in the Methods section.

Page 5, Lines 69–73

This study had a prospective cohort design. This study was approved by the ethics committee of Suiseikai Kajikawa Hospital and was performed according to the national government's guidelines based on the Helsinki Declaration of 1964. Written informed consent was obtained from all the patients or their relatives. All assessors were blinded to the data for analyses.

Page 5, Line 76–Page 6, Line 86

All consecutive patients with acute ischemic or hemorrhagic stroke who were admitted to Suiseikai Kajikawa Hospital within 1 week of disease onset between June 1, 2015, and August 31, 2015 were included in this cohort study. Patients who were under 20 years old, did not provide consent (for patients who could not provide consent, consent was obtained from their relatives), were in coma (best eye response score on the Glasgow coma scale of 1), underwent craniotomy, were on mechanical ventilation, or diagnosed with aspiration pneumonia in the hospital were excluded. Aspiration pneumonia was diagnosed according to the criteria of the Centers for Disease Control and Prevention [10]. We monitored patients for signs of pneumonia for 30 days after admission. If the patients were discharged before the 30th day, then they were monitored until the day of discharge.

Comment 2: Please add explanation on sample size determination; the parameters used in the calculation.

Response 2: We calculated the sample size using data from our previous report. In the previous report, we measured the tongue pressure in patients with acute stroke and compared the modified Mann Assessment of Swallowing Ability (MASA) score, which is an established bedside assessment tool that indicates the risk of swallowing dysfunction. It is reported that a modified MASA score <95 suggests swallowing dysfunction. In the previous study, we investigated the difference in tongue pressure between the modified MASA score <95 and ≥95 groups; the effect size was 1.56 (strong). This parameter was used in calculating the sample size for this study. This explanation has been added to the Methods section.

Page 9, Lines 138–146

We calculated the required sample size based on our previous study on tongue pressure in acute stroke [7]. In the previous study, tongue pressure was compared using the modified Mann Assessment of Swallowing Ability (MASA) score, which is an established bedside assessment tool that indicates the risk of developing swallowing dysfunction. A modified MASA score of <95 suggests swallowing dysfunction. The difference in the tongue pressure between the modified MASA score <95 and ≥95 groups was investigated, and the effect size was strong. This parameter was used to calculate the study sample size. Using an alpha level of 0.05 and a power of 0.80, 56 participants were required to participate in this study.

Comment 3: Please explain on sampling method in related to “consecutive acute stroke patients…”.

Response 3: We revised the information on the sampling method accordingly. Following the inclusion and exclusion criteria, 80 patients were included in the study finally. We revised the manuscript and added the flowchart of the inclusion and exclusion criteria in the Results section.

Page 5, Line 76–Page 6, Line 86

All consecutive patients with acute ischemic or hemorrhagic stroke who were admitted to Suiseikai Kajikawa Hospital within 1 week of disease onset between June 1, 2015, and August 31, 2015 were included in this cohort study. Patients who were under 20 years old, did not provide consent (for patients who could not provide consent, consent was obtained from their relatives), were in coma (best eye response score on the Glasgow coma scale of 1), underwent craniotomy, were on mechanical ventilation, or diagnosed with aspiration pneumonia in the hospital were excluded. Aspiration pneumonia was diagnosed according to the criteria of the Centers for Disease Control and Prevention [10]. We monitored patients for signs of pneumonia for 30 days after admission. If the patients were discharged before the 30th day, then they were monitored until the day of discharge.

Page 10, Lines 159–165

A total of 105 patients were eligible for this study. No patient was below 20 years old. Eleven patients or their relatives did not provide consent, two patients underwent craniotomy, and one patient was on mechanical ventilation. Among the remaining patients, eight patients were not evaluated because of a poor general condition, and three patients were diagnosed with aspiration pneumonia. Therefore, 80 patients were included in the study. A flowchart of the inclusion and exclusion criteria is shown in Fig 1.

Comment 4: Should report median with interquartile range.

Response 4: We have added the interquartile range in the Manuscript and tables.

Page 9, Lines 134–137

Data are expressed as median (minimum, 25th percentile, 75th percentile, maximum) for continuous variables (age, body mass index, NIHSS score, FILS, diet form, and tongue pressure) and as frequencies (percentages) for categorical variables (stroke subtypes and histories of comorbidities).

Comment 5: Revise the findings. Should state the outcome for each analysis. Tongue pressure influence the dysphagia diet or vice versa. Please explain.

Response 5: Thank you for this recommendation. Tongue pressure influences the dysphagia diet form. Thus, tongue pressure should be an explanatory variable, and the dysphagia diet form should be the response variable. We have deleted FILS and the dysphagia diet form in Table 2. We also performed a reanalysis with FILS and dysphagia diet form as response variables (Table 3). We revised the tables and manuscript as follows. The Abstract was also revised.

Page 2, Lines 28–30

In multivariate analysis, only tongue pressure was significantly associated with the dysphagia diet form (p<0.001).

Page 9, Lines 146–150

To evaluate the association between tongue pressure and dysphagia diet form, the baseline data of the patients were analyzed, and two-step strategies were used to assess the significant importance of variables in the association between tongue pressure and FILS/dysphagia diet form using a least square linear regression analysis.

Page 11, Line 180–Page 12, Line 185

The associations between the factors listed in Table 1 (except for FILS and dysphagia diet form because they are response variables) and tongue pressure at admission were evaluated. In univariate analysis, age, sex, body mass index, diabetes mellitus, dyslipidemia, and NIHSS score were associated with tongue pressure. In multivariate analysis, age and NIHSS score were significant independent factors for tongue pressure (Table 2). 

Page 12, Line 195–Page 13, Line 202

Next, the associations between the factors listed in Table 1 and FILS/dysphagia diet form were evaluated. Regarding FILS in univariate analysis, age, sex, dyslipidemia, NIHSS score, and tongue pressure were associated with FILS. In multivariate analysis, NIHSS score and tongue pressure were significant independent factors for FILS (Table 3A). Regarding dysphagia diet form, in univariate analysis, age, sex, body mass index, NIHSS score, and tongue pressure were associated with dysphagia diet form. In multivariate analysis, only tongue pressure was significantly associated with the dysphagia diet form (Table 3B). 

Comment 6: The authors should control the confounder for comparison of mean tongue pressure for each group of dysphagia diet; ANCOVA instead of ANOVA.

Response 6: We re-analyzed using ANCOVA. We revised the Methods section.

Page 10, Lines 153–154

The analysis of co-variance was used to compare the mean tongue pressure between the dysphagia diet groups.

Page 14, Lines 214–216

The mean tongue pressures of the dysphagia diet groups were compared (Fig. 2). After adjusting for age and NIHSS score, which were associated with tongue pressure, the mean tongue pressure increased significantly as the dysphagia diet code increased.

Comment 7: Do all the assumptions of the test fulfilled? Please explain.

Response 7: As mentioned in “Response 5,” we reanalyzed the data and specified which variables were the response and explanatory variables. Based on those, we performed multivariate and ROC analyses. We have clearly presented the assumptions and made revisions to explain this in the revised manuscript.

Page 5, Lines 63–65

We investigated the association between tongue pressure and dysphagia diet form and searched for the suitable tongue pressure for choosing each diet form.

Page 9, Line 146–Page 10, Line 156

To evaluate the association between tongue pressure and dysphagia diet form, the baseline data of the patients were analyzed, and two-step strategies were used to assess the significant importance of variables in the association between tongue pressure and FILS/dysphagia diet form using a least square linear regression analysis. First, a univariate analysis was performed, and factors with p <0.05 were selected. In a multi-factorial analysis, least linear regression analyses were performed with the selected factors. The analysis of co-variance was used to compare the mean tongue pressure between the dysphagia diet groups. Moreover, receiver operating characteristic (ROC) analyses were performed to determine the tongue pressure to predict suitable diet forms. Statistical significance was set at p <0.05.

Comment 8: Please justify the use of ROC analysis.

Response 8: As we mentioned in “Response 5,” FILS and the dysphagia diet form are response variables. Because we revealed the association between FILS/dysphagia diet form and tongue pressure (Table 3), an ROC analysis was performed to determine the tongue pressure that could predict the dysphagia diet form.

Page 15, Lines 222–224

Because we revealed the association between FILS/dysphagia diet form and tongue pressure (Table 3), ROC analyses were performed to determine the tongue pressure that can be used to predict a suitable dysphagia diet.

Comment 9: Table 1: Delete All. Should write n=80 at the end of table title. Replace it with n(%). Delete n(%) written after factors. Should report median (IQR).

Response 9: We revised Table 1 as recommended.

Comment 10: Table 2: Any reason run different model. What is predictor? Does it refer to regression coefficient? Please the results of categorical factors. Should state the reference.

Response 10: We deleted FILS and the dysphagia diet form in Table 2 because they should be response variables. Thus, the different models were also deleted. Predictor means regression coefficient. Regarding categorical factors, we have stated the reference in the footnote of the tables (Revised Tables 2 and 3).

Reviewer #2: This study by Nakamori et al evaluated if tongue pressure measurement is a sensitive indicator for determining diet form and swallowing capability among a sample of 80 patients hospitalized for acute stroke in a Japanese hospital. The authors found that tongue pressure measured on admission by a multidisciplinary team is significantly associated with diet form and swallowing. The manuscript is very well-written and describes an interesting research question with clear clinical implications. I have some comments about the presentation of the methods and results:

Comment 1: Page 10 lines 144-146: Please describe the additional parameters used to calculate sample size (e.g., effect size based on prior literature)

Response 1: We calculated the sample size using data from our previous report. In the previous report, we measured the tongue pressure in patients with acute stroke and compared the modified Mann Assessment of Swallowing Ability (MASA) score, which is an established bedside assessment tool that indicates the risk of swallowing dysfunction. It is reported that a modified MASA score of <95 suggests swallowing dysfunction. In the previous study, we investigated the difference in tongue pressure between the modified MASA score <95 and ≥95 groups; the effect size was 1.56 (strong). This parameter was used for calculating the sample size for this study. This explanation has been added to the Methods section.

Page 9, Lines 138–146

We calculated the required sample size based on our previous study on tongue pressure in acute stroke [7]. In the previous study, tongue pressure was compared using the modified Mann Assessment of Swallowing Ability (MASA) score, which is an established bedside assessment tool that indicates the risk of developing swallowing dysfunction. A modified MASA score of <95 suggests swallowing dysfunction. The difference in the tongue pressure between the modified MASA score <95 and ≥95 groups was investigated, and the effect size was strong. This parameter was used to calculate the study sample size. Using an alpha level of 0.05 and a power of 0.80, 56 participants were required to participate in this study.

Comment 2: Page 10 lines 148: More details about the multivariable model are important to state. Please explicitly state what the dependent variable is and how it was operationalized (continuous, categorical, etc). Along the same lines, what type of multivariable model was used (linear, logistic, etc)? What covariates did you consider in the model and how were they operationalized (Was NIHSS at admission and was it measured as a continuous variable? Same with BMI and comorbidities.)

Response 2: Thank you for your recommendation. In the previous manuscript, there was some ambiguity related to the explanatory and response variables. The FILS and dysphagia diet form should be response variables; however, we analyzed them as explanatory variables in Table 2. Thus, we deleted FILS and the dysphagia diet form in Table 2. We also performed a reanalysis using FILS and dysphagia diet form as response variables (Table 3). We revised the tables and manuscript accordingly. We used age, body mass index, NIHSS score, FILS, diet form, and tongue pressure as continuous variables, whereas stroke subtypes and comorbidities were categorical variables. The multivariable models were least linear regressions. We have added this information in the Methods section.

Page 9, Lines 134–137

Data are expressed as median (minimum, 25th percentile, 75th percentile, maximum) for continuous variables (age, body mass index, NIHSS score, FILS, diet form, and tongue pressure) and as frequencies (percentages) for categorical variables (stroke subtypes and histories of comorbidities).

Page 9, Lines 146–150

To evaluate the association between tongue pressure and dysphagia diet form, the baseline data of the patients were analyzed, and two-step strategies were used to assess the significant importance of variables in the association between tongue pressure and FILS/dysphagia diet form using a least square linear regression analysis.

Page 10, Line 150–Page 10, Line 153

First, a univariate analysis was performed, and factors with p <0.05 were selected. In a multi-factorial analysis, least linear regression analyses were performed with the selected factors.

Page 11, Line 180–Page 12, Line 185

The associations between the factors listed in Table 1 (except for FILS and dysphagia diet form since they are response variables) and tongue pressure at admission were evaluated. In univariate analysis, age, sex, body mass index, diabetes mellitus, dyslipidemia, and NIHSS score were associated with tongue pressure. In multivariate analysis, age and NIHSS score were significant independent factors for tongue pressure (Table 2). 

Page 12, Line 195–Page 13, Line 202

Next, the associations between the factors listed in Table 1 and FILS/dysphagia diet form were evaluated. Regarding FILS in univariate analysis, age, sex, dyslipidemia, NIHSS score, and tongue pressure were associated with FILS. In multivariate analysis, NIHSS score and tongue pressure were significant independent factors for FILS (Table 3A). Regarding dysphagia diet form, in univariate analysis, age, sex, body mass index, NIHSS score, and tongue pressure were associated with dysphagia diet form. In multivariate analysis, only tongue pressure was significantly associated with the dysphagia diet form (Table 3B). 

Comment 3: Page 11 (Table 1): It would be more helpful to know the IQR when you present descriptive statistics (rather than minimum and maximum), especially for your key variables of interest (FILS, diet form). More information about distribution of the FILS, diet form, and tongue pressure, would be helpful to the reader since these are key variables in your study.

Response 3: We have added the interquartile range in the manuscript and tables. We have submitted all other relevant data of the study as supplemental data.

Page 9, Lines 134–137

Data are expressed as median (minimum, 25th percentile, 75th percentile, maximum) for continuous variables (age, body mass index, NIHSS score, FILS, diet form, and tongue pressure) and as frequencies (percentages) for categorical variables (stroke subtypes and histories of comorbidities).

Comment 4: Page 13 (Table 2): Some comments on the presentation of data in Table 2. What is the reference group for categorical variables (sex, hypertension, diabetes, etc)? This should be clearly labeled in the table. Is the “predictor” column referring to beta coefficients?

Response 4: First, as Reviewer 1 recommended, we have deleted FILS and the dysphagia diet form in Table 2 because they should be response variables. Thus, the different models were also deleted. Predictor means regression coefficient. Regarding categorical factors, we have stated the reference in the footnote of the tables (Revised Tables 2 and 3).

Comment 5: Page 15 Figure 1: Please indicate that the bars are (95%?) confidence intervals

Response 5: We have added the explanation of the bars in the footnotes.

Comment 6: Page 15 lines 191-192: The mention of ROC analyses was a surprise as I was reading the article. ROC needs to be described in the methods section.

Response 6: We have details of the ROC analyses in the Methods and Results sections.

Page 10, Lines 154–155

Moreover, receiver operating characteristic (ROC) analyses were performed to determine the tongue pressure to predict suitable diet forms.

Comment 7: I’m a little confused about the wording in the manuscript. Are you classifying FILS (swallowing capability) as another measure of diet form? For example, the caption for figure 2 only refers to diet form, even though panel A discusses FILS.

Response 7: FILS measures swallowing ability without considering the diet form. However, in a real clinical setting, it is important to know whether the patient can take food orally completely, which corresponds to a FILS of 7. We have explained this carefully in the manuscript for readers to understand. 

Page 15, Lines 222–229

Because we revealed the association between FILS/dysphagia diet form and tongue pressure (Table 3), ROC analyses were performed to determine the tongue pressure that can be used to predict a suitable dysphagia diet. FILS measures swallowing ability without considering the diet form. However, in a real clinical setting, it is important to know whether the patient can take food completely orally. A state of complete oral intake corresponds to FILS ≥7; therefore, we investigated the optimal cutoff tongue pressure for predicting a FILS of ≥7. The cut-off was 9.6 kPa (χ2=46.83, p<0.001, sensitivity 100.0%, specificity 92.8%, AUC=0.991) (Fig 2A).

Comment 8: It may be useful to have information about training to conduct tongue pressure. Who performed it in the multidisciplinary team? Was there standardized training or is this part of routine clinical practice so this type of standardization is not needed? This would be useful if the reader wants to consider how they can translate these results to their own clinical practice.

Response 8: Tongue pressure measurement is part of the routine clinical practice in our institution, and it does not require special skills. However, since standardization is essential for any clinical study, the nurses were trained. Moreover, we reconfirmed the reliability of tongue pressure measurement. We have added this explanation in the revised manuscript. 

Page 8, Lines 128–131

The reliability of intraindividual measurements has been reported previously [17]. We reconfirmed the reliability of the measurements in this study. Measurements were taken repeatedly for ten days in normal subjects, and the resulting coefficient of variation was 5.64%.

Page 18, Lines 289–293

In clinical practice, it is important for a multidisciplinary swallowing team to diagnose dysphagia as early as possible to start managing nutrition and preventing aspiration pneumonia. Although we reconfirmed the reliability of tongue pressure measurement, it does not require special skills. In this study, tongue pressure was measured on admission—this is recommendable.

Reviewer #3: Thank you for allowing us to read your interesting manuscript. This study provides information about the tongue pressure, such as a non-invasive bedside assessment tool for deciding the diet form for acute stroke patients. It is a great effort, congratulations. Please consider the following comments.

Comment 1: In many parts of the manuscript appears the word “dysphasia” (9 times) including the tittle, which I think should be corrected to “dysphagia”.

Response 1: Thank you for pointing out this error. We have corrected the word “dysphasia” to “dysphagia”.

Comment 2: It should be important that the abstract conclusion make it clear that tongue pressure is just one tool within other evaluations (such cough function) that must be made to patient to decide the type of diet, similar to the idea expressed in the extend conclusion.

Response 2: Thank you for this recommendation. We have revised the conclusion of the Abstract accordingly.

Page 2, Line 34–Page 3, Line 37

Tongue pressure is one of the sensitive indicators for choosing dysphagia diet forms in patients with acute stroke. A combination of simple modalities will increase the accuracy of the swallowing assessment and choice of the diet form.

Comment 3: More than 50% of the sample were minor strokes (NIHSS �5), which could be a limitation when extending the results to more serious strokes.

Response 3: Thank you for this recommendation. We added this as a limitation.

Page 19, Lines 294–299

This study had limitations. First, it was performed in a single institution and might have sources of bias. The sample size was small. More than half of the patients had mild stroke, which could prevent the generalizability of the results to a population with severe stroke. It would be difficult to measure the tongue pressure of patients with severe stroke accurately. Future multicenter research would be needed to eliminate the effects of bias.

Comment 4: It should be important that you give a recommendation regarding the appropriate timing for the initiation of this post-stroke test (tongue pressure), depending on the moment in which you carried out these evaluations in the study.

Response 4: It is important to evaluate dysphagia as early as possible to start managing nutrition and preventing aspiration pneumonia. In this study, we measured the tongue pressure on admission. We have added this recommendation in the Discussion section.

Page 18, Lines 289–293

In clinical practice, it is important for a multidisciplinary swallowing team to diagnose dysphagia as early as possible to start managing nutrition and preventing aspiration pneumonia. Although we reconfirmed the reliability of tongue pressure measurement, it does not require special skills. In this study, tongue pressure was measured on admission—this is recommendable.

---

## [Decision Letter · Decision Letter 1]

24 May 2021

Relationship between tongue pressure and dysphagia diet in patients with acute stroke

PONE-D-21-05820R1

Dear Dr. Masahiro Nakamori,

We’re pleased to inform you that your manuscript has been judged scientifically suitable for publication and will be formally accepted for publication once it meets all outstanding technical requirements.

Kind regards,

Wisit Cheungpasitporn, MD

Academic Editor

PLOS ONE

Additional Editor Comments:

I reviewed the revised manuscript and the response to reviewers' comments. Revised Manuscript is well written. All comments have been addressed and thus accepted for publication.

Reviewers' comments:

Reviewer's Responses to Questions

**Comments to the Author**

1. If the authors have adequately addressed your comments raised in a previous round of review and you feel that this manuscript is now acceptable for publication, you may indicate that here to bypass the “Comments to the Author” section, enter your conflict of interest statement in the “Confidential to Editor” section, and submit your "Accept" recommendation.

Reviewer #2: All comments have been addressed

Reviewer #3: All comments have been addressed

2. Is the manuscript technically sound, and do the data support the conclusions?

Reviewer #2: (No Response)

Reviewer #3: Yes

3. Has the statistical analysis been performed appropriately and rigorously? 

Reviewer #2: (No Response)

Reviewer #3: Yes

4. Have the authors made all data underlying the findings in their manuscript fully available?

Reviewer #2: (No Response)

Reviewer #3: Yes

5. Is the manuscript presented in an intelligible fashion and written in standard English?

Reviewer #2: (No Response)

Reviewer #3: Yes

6. Review Comments to the Author

Reviewer #2: (No Response)

Reviewer #3: Dr Masahiro Nakamori, thank you for making the requested changes, the article is much clearer now and I consider it is a good contribution for this subtype of patients.

7. PLOS authors have the option to publish the peer review history of their article (what does this mean?). If published, this will include your full peer review and any attached files.

Reviewer #2: No

Reviewer #3: **Yes: **Vanessa Cano-Nigenda

---

## [Editor Report · Acceptance letter]

26 May 2021

PONE-D-21-05820R1 

Relationship between tongue pressure and dysphagia diet in patients with acute stroke 

Dear Dr. Nakamori:

I'm pleased to inform you that your manuscript has been deemed suitable for publication in PLOS ONE. Congratulations! Your manuscript is now with our production department. 

Kind regards, 

on behalf of

Dr. Wisit Cheungpasitporn 

Academic Editor

PLOS ONE